# Age, sex, and other demographic trends in sexual behavior in the United States: Initial findings of the sexual behaviors, internet use, and psychological adjustment survey

**Hannah Roberts**[ID]*, **Angus Clark, Carter Sherman**[ID], **Mary M. Heitzeg, Brian M. Hicks**\*

Department of Psychiatry, University of Michigan, Ann Arbor, MI, United States of America

\* rohannah@med.umich.edu (HR); brianhic@med.umich.edu (BMH)

## Abstract

It remains unclear how the seemingly ubiquitous use of the internet impacts user's offline personal relationships, particularly those that are romantic or sexual. Therefore, we conducted a national online survey to better understand the associations among internet use, sexual behavior, and adjustment called the Sexual Behaviors, Internet Use, and Psychological Adjustment Survey (SIPS). Here, we report patterns of sexual behavior in a sample of adults ($N$ = 1987; ages 18–70) in the United States to establish its representativeness and consistency with similar recent surveys. We found age- and sex-related trends in oral, vaginal, and anal sex in terms of prevalence, frequency, number of partners, and age of initiation consistent with prior studies. We also detected differences in sexual behaviors based on relationship status and sexual orientation, but small and relatively few significant differences across racial and ethnic groups. The results confirm and expand upon trends identified in prior national surveys of sexual behavior, establishing the representativeness of the SIPS sample for use in future research examining the links among sexual behaviors and romantic relationships, internet use, and adjustment.

## Introduction

The encroachment of the internet into everyday life has changed the way people form and maintain romantic and sexual relationships. For example, dating apps (e.g., Tinder, Bumble, Hinge) and social media networks (e.g., Facebook, Twitter, Instagram) welcome millions of users a day, yet the impacts of technology use on in-person relationships and sexual activity remains a relatively new area of study [1–5]. To better understand these novel associations and whether internet technology has altered patterns of coupling behavior, we conducted a national online survey called the Sexual Behaviors, Internet Use, and Psychological Adjustment Survey (SIPS), which assessed sexual behaviors, social media and dating app use, and several domains of adjustment including mental health, substance use, and interpersonal functioning.

**Data Availability Statement:** Data are hosted through the Open Science Framework (OSF), and

can be accessed using the following link: https://osf.io/g8phr/.

**Funding:** This work was supported by United States Public Health Service grants R01 AA024433 (Hicks), R01 AA02579 (Heitzeg), and T32 AA007477 (Blow) from the National Institute on Alcohol Abuse and Alcoholism.

**Competing interests:** The authors have declared that no competing interests exist.

An important first step to making valid inferences about the associations among these social and sexual domains is to describe normative patterns of sexual behavior. It is especially important to document patterns of sexual behavior in demographically diverse samples with varying ages, genders, incomes, races, ethnicities, and sexual orientations represented. Here, we utilize the SIPS sample to characterize demographic trends in multiple sexual behaviors. We also sought to replicate and expand upon recent surveys that reported nationally representative demographic trends in sexual behavior to validate the representativeness of the SIPS sample and bolster the reliability and validity of past estimates of sexual behavior. Given the shift towards online data collection in recent years, it is especially critical to replicate findings that utilize relatively novel, modern sampling methods, i.e., internet-based national surveys. Thus, as participation in internet-based social networking continues to rise, it is critical to understand if and how these social trends are impacting in-person social relationships, sexual behaviors, and well-being more broadly.

## Surveys of sexual behavior in the U.S.

The taboo nature of sex has historically proven to be a barrier in ascertaining demographically representative surveys of sexual behavior. In the mid-20th century, Kinsey and colleagues [6, 7] conducted a series of landmark studies on sexual activity in men and women. However, these studies emphasized sexual diversity rather than estimates of population prevalence and failed to assess how sexual diversity presents across different groups (e.g., sexual orientation, race) [6, 7]. Towards the end of the 20th century, several demographically representative surveys emerged to help understand trends in common and uncommon (e.g., "kink" behavior) sexual behaviors, as well as sexual health behaviors (e.g., use of contraception, risk for sexually transmitted disease) [8–11]. Though these surveys were robust and collectively add considerable information about sexual behavior across age and sex, they were mainly conducted through in-person interview and questionnaire procedures.

The recent advent of the internet and online surveys has afforded new opportunities for more easily and efficiently recruiting nationally representative samples and targeted samples whose members constitute a sexual minority in the broader population. Recently, Herbenick et al. [12] collected a nationally representative online sample of U.S. adults to estimate the prevalence of oral sex, vaginal sex, anal sex, and a variety of other sexual behaviors. Most respondents had engaged in oral (84.0%) and vaginal (88.5%) sex during their lifetime, but significantly fewer people had ever engaged in anal sex (33.2%). Women reported higher rates of lifetime vaginal sex (91.1%) and receptive anal sex (37.3%) than men (85.6% and 9.3%, respectively), but 42.6% of men reported lifetime insertive anal sex. Rates of vaginal, oral, and anal sex in the past 12 months and past 30 days exhibited a quadratic, inverted U-shape with age such that rates were highest for people in young (ages 25–29 years old) to middle adulthood (ages 30–49 years old) and lower at younger (ages 18–24 years old) and older ages (> 50 years old).

Although these trends are informative, it is also important to consider how various sexual behaviors differ across other relevant variables such as relationship status, sexual orientation, race, and ethnicity. For example, 4–8% of individuals report participation in non-monogamous relationships, allowing for the possibility of multiple sexual and romantic partners [13]. Cohabitation can also affect the accessibility of sexual partners. Indeed, married couples who are cohabitating are almost twice as likely to report sexual intercourse over the past 90 days compared to those who are not cohabitating [14–16]. Further, homosexual and bisexual men reported lower rates of vaginal sex, but higher rates of insertive and receptive anal sex compared to heterosexual men [17].

## Current study

We recently conducted the SIPS, an online survey of 1,987 U.S. adults to help answer questions about the associations between internet use, sexual behavior, and psychosocial adjustment. Questions regarding sexual behavior focused specifically on single-partner oral, vaginal, and anal sex. In addition to prevalence, we also collected details about frequency, age of initiation, and number of sexual partners. We first examined the rates and demographic trends in sexual behaviors in the SIPS sample and then compared our findings with prior estimates from similar surveys to evaluate the representativeness of the current sample. Expected findings related to age and sex trends in sexual behavior included:

1. The lifetime prevalence of oral and vaginal sex (>80%) would be higher than for anal sex (<40%) [12, 18].

2. The prevalence and frequency of oral, vaginal, and anal sex in the past 12 months and past 30 days would exhibit a quadratic association with age, such that rates of sexual behavior would be highest in young (ages 25–29) and middle adulthood (ages 30–49), and lower in emerging (ages 18–24) and older adulthood (ages 50 and older) [12].

3. The age of initiation for oral and vaginal sex would be younger than the age of initiation for anal sex [18].

4. Males would report more sexual partners than females [10, 19, 20].

Beyond age and gender trends, we also sought to replicate several findings regarding differences in sexual behavior related to relationship status, sexual orientation, race, and ethnicity. Expected findings in these domains included:

5. People in a serious relationship would report engaging in more sexual behaviors than people not in a relationship [14–16].

6. Heterosexual and bisexual participants would report higher rates of vaginal sex than homosexual participants, and homosexual males would report higher rates of anal sex than heterosexual males [17].

7. Black participants would report an earlier age of initiation and more sexual partners than White and Asian participants [14, 15, 21–23].

8. Hispanic participants would report an older age of initiation and fewer sexual partners than non-Hispanic participants [14, 15].

## Method

### Recruitment

Data were collected in February and March 2020 using an actively managed, double-opt-in research panel in the Qualtrics XM survey software. Recruitment procedures were intended to capture a sample consistent with the demographics of the United States general population in terms of age, gender, education, race, and household income. Quotas were created for each demographic variable and monitored while the survey was in the field. Respondents were recruited using a dashboard-style web page on the Qualtrics website and cellphone app where participants saw a list of surveys that they had the option to participate in. Recruitment was also conducted via emails sent to established panel members within the Qualtrics database. In all recruitment methods, potential participants received information regarding the estimated

length of the survey and the compensation rate for completing it. Specific details about the survey content were not available until the participants opted-in to deter self-selection bias.

Upon opting into the study, participants read and electronically signed a consent form containing an overview of the survey contents. Participation was voluntary and anonymous as no individually identifying information was collected. Contact information for the research team was provided if participants had questions about the survey. The median response time for completing the survey was 23.1 minutes. The University of Michigan Medical School Institutional Review Board (IRB) reviewed all study protocols (HUM00170909).

### Sample characteristics

The survey was completed by 2100 respondents. Data were manually checked and 113 respondents were excluded due to multiple inconsistent and unreasonable answers (e.g., age of sexual initiation was older than current age). Single values were excluded on a case-by-case basis if all other responses from that participant were within a plausible range of values, assuming the respondent misread or misunderstood a single question.

### Measures

**Demographics.**   We asked respondents to report their current age, biological sex, gender, race, ethnicity, and sexual orientation. Table 1 provides descriptive statistics for these and other variables.

**Relationship status.**   We asked respondents to report their current romantic relationship status. Although we provided 11 possible response options, these options were collapsed into three categories for analysis. Respondents were able to select multiple response options if applicable. *Not dating or not in a relationship* included the responses: *rarely date* and *not dating now*. Being in a *casual relationship* included the responses: *mostly going out with one person and dating a few others*, *dating or seeing more than one person*, and *dating or seeing one person casually*. Being in a *serious relationship* included the responses: *married; living with someone; engaged; planning to get engaged, married, or live together; in a serious relationship;* and *have a monogamous boyfriend/girlfriend*.

**Oral, vaginal, and anal sex.**   First, we asked respondents to report if they had ever engaged in oral, vaginal, and anal sex (*yes/no*) to assess prevalence rates for each behavior. Oral sex was defined as when "a person puts their mouth on another person's sex organs" and vaginal sex was defined as "inserting the penis into the vagina". Anal sex was described as when "a man inserts his penis into his partner's anus or asshole". If a respondent reported ever engaging in a given sexual behavior, they were then asked to indicate how frequently they had engaged in that behavior during the past 12 months and the past 30 days *(not at all, less than once a month, once a month, multiple times per month, once a week, multiple times per week, every day, multiple times per day)*. Next, we asked respondents how many partners they had engaged in oral, vaginal, and anal sex with during their lifetime and in the past 12 months. Finally, respondents were asked to indicate the age at which they first engaged in oral, vaginal, and anal sex.

### Data analysis

Logistic and linear regression models were used to analyze the demographic trends associated with each sexual behavior. For each model, we entered Age, $Age^2$, biological Sex, an Age x Sex interaction, and an $Age^2$ x Sex interaction as predictor variables of each sexual behavior outcome and set an alpha of $p < .005$ for statistical significance. Age was grand mean centered before calculating the interaction terms. Although the descriptive tables and plots are

**Table 1. Demographic characteristics.**

| | Total (*N* = 1,987) | Men (*n* = 953) | Women (*n* = 1015) |
|---|---|---|---|
| | % (n) | % (n) | % (n) |
| **Age (yrs)** | | | |
| 18–24 | 13.6 (270) | 8.9 (85) | 17.2 (175) |
| 25–29 | 9.5 (189) | 6.6 (63) | 11.9 (121) |
| 30–39 | 19.8 (393) | 16.2 (154) | 23.4 (237) |
| 40–49 | 17.0 (338) | 16.3 (155) | 18.0 (183) |
| 50–59 | 16.2 (321) | 17.8 (170) | 14.8 (150) |
| 60–70 | 24.0 (476) | 31.7 (302) | 14.7 (149) |
| **Race** | | | |
| White | 78.9 (1568) | 81.0 (772) | 77.3 (785) |
| Black | 13.0 (259) | 12.5 (119) | 13.4 (136) |
| Native American | 0.9 (18) | 0.6 (6) | 1.0 (10) |
| Asian | 4.9 (97) | 4.3 (41) | 5.3 (54) |
| Native Hawaiian | 0.3 (6) | 0.1 (1) | 0.5 (5) |
| Don't know | 0.6 (11) | 0.2 (2) | 0.9 (9) |
| Some other race | 4.0 (80) | 2.6 (25) | 5.3 (54) |
| Hispanic- any race | 17.1 (340) | 10.9 (104) | 22.8 (231) |
| **Education** | | | |
| Less than 7th grade | 0.2 (4) | 0.1 (1) | 0.2 (2) |
| Junior high school | 0.5 (9) | 0.2 (2) | 0.7 (7) |
| Some high school, no degree | 4.2 (84) | 3.3 (31) | 5.0 (51) |
| High school diploma/ GED | 25.5 (506) | 22.8 (217) | 28.2 (286) |
| Vocational tech diploma | 5.1 (101) | 6.0 (57) | 4.1 (42) |
| Some college, no degree | 23.1 (459) | 23.9 (228) | 22.1 (224) |
| Associates degree | 9.1 (180) | 7.8 (74) | 10.3 (105) |
| Bachelor's degree | 20.7 (412) | 23.4 (223) | 18.4 (187) |
| Master's degree | 9.1 (181) | 9.7 (92) | 8.7 (88) |
| Doctorate | 2.6 (51) | 2.9 (28) | 2.3 (23) |
| **Sexual Orientation** | | | |
| 100% Heterosexual | 82.4 (1638) | 86.9 (828) | 79.5 (807) |
| Mostly heterosexual | 6.9 (137) | 3.9 (37) | 9.7 (98) |
| Bisexual | 6.0 (119) | 3.9 (37) | 7.6 (77) |
| Mostly homosexual | 1.2 (23) | 0.8 (8) | 1.2 (12) |
| 100% Homosexual | 3.0 (60) | 4.3 (41) | 1.5 (15) |
| Questioning | 0.2 (3) | 0 (0) | 0.2 (2) |
| Asexual | 0.4 (7) | 0.2 (2) | 0.4 (4) |
| **Marital Status** | | | |
| Married | 48.6 (965) | 55.5 (529) | 42.7 (433) |
| Widowed | 2.9 (58) | 2.2 (21) | 3.3 (34) |
| Divorced | 12.3 (245) | 10.1 (96) | 14.7 (149) |
| Separated | 1.6 (31) | 1.2 (11) | 2.0 (20) |
| Never married | 34.6 (688) | 31.1 (296) | 37.3 (379) |
| **Relationship Status** | | | |
| Married | 47.3 (940) | 54.6 (520) | 41.1 (417) |
| Living with someone | 9.3 (185) | 7.6 (72) | 10.8 (110) |
| Engaged | 3.0 (60) | 1.4 (13) | 4.4 (45) |
| Planning to get engaged/ married/live together | 2.5 (49) | 1.7 (16) | 3.3 (33) |

(*Continued*)

**Table 1.** (Continued)

|  | Total (*N* = 1,987) | Men (*n* = 953) | Women (*n* = 1015) |
|---|---|---|---|
|  | % (n) | % (n) | % (n) |
| In a serious relationship with one person | 8.7 (173) | 6.4 (61) | 10.6 (108) |
| Have a boy/girlfriend and only see them | 3.0 (59) | 2.3 (22) | 3.5 (36) |
| Mostly going out with one person and dating a few others | 0.8 (15) | 0.3 (3) | 1.2 (12) |
| Dating or seeing more than one person | 1.8 (35) | 1.8 (17) | 1.7 (17) |
| Dating or seeing one person casually | 2.7 (53) | 2.0 (19) | 3.2 (32) |
| Rarely date | 3.8 (75) | 3.7 (35) | 3.9 (40) |
| Not dating now | 24.3 (483) | 23.3 (222) | 25.0 (254) |
| **Annual Household Income** |  |  |  |
| $0–9,999 | 8.5 (169) | 5.5 (52) | 11.2 (114) |
| $10,000-$24,999 | 11.7 (232) | 7.1 (68) | 15.7 (159) |
| $25,000–49,999 | 24.8 (492) | 17.7 (169) | 31.2 (317) |
| $50,000–74,999 | 20.7 (411) | 26.9 (256) | 14.9 (151) |
| $75,000–99,999 | 12.2 (243) | 19.5 (186) | 5.5 (56) |
| $100,000–149,999 | 10.2 (203) | 13.4 (128) | 7.4 (75) |
| $150,000–199,999 | 6.1 (121) | 4.7 (45) | 7.5 (76) |
| $200,000+ | 5.8 (116) | 5.1 (49) | 6.6 (67) |
| **Employment Status** |  |  |  |
| Full-time | 43.0 (855) | 51.0 (486) | 24.2 (365) |
| Part-time | 14.0 (278) | 10.1 (96) | 17.4 (177) |
| Casual/seasonal | 0.8 (16) | 1.0 (10) | 0.4 (4) |
| Volunteer | 1.0 (20) | 0.8 (8) | 1.2 (12) |
| Not working (temporary layoff) | 0.8 (16) | 1.2 (11) | 0.5 (5) |
| Not working (looking for a job) | 5.6 (111) | 3.8 (36) | 7.2 (73) |
| Not working (retired) | 15.5 (307) | 22.2 (212) | 9.4 (95) |
| Not working (disabled) | 6.0 (120) | 4.6 (44) | 7.4 (75) |
| Not working (student) | 4.2 (83) | 2.7 (26) | 5.2 (53) |
| Not working (homemaker) | 7.7 (153) | 0.9 (9) | 14.1 (143) |
| Not working (other) | 1.4 (28) | 1.6 (15) | 1.3 (13) |

**Note.** Additional responses for gender identity included: 0.3% (5) Trans male/trans man, 0.1% (2) Trans female/trans woman, 0.4% (8) Gender Queer/Gender Non-Conforming, 0.2% (3) Different identity, and 0.1% (1) Refuse to answer.

organized based on age bins, the continuous age variables were used in all statistical analyses. Extreme values to free-response questions (~1%) for the number of partners for oral, vaginal, and anal sex for lifetime and the past 12 months were winsorized.

One-way ANOVAs were then used to test for mean differences in the frequency of sexual behaviors (past 12 months and 30 days), the number of sexual partners (lifetime and past 12 months), and age at sexual initiation across relationship status, sexual orientation, race, and ethnicity. Post hoc comparisons using the Tukey HSD test ($\alpha$ = .005) were used to test for significant differences among respondents either in a serious relationship, casually dating, or not currently dating or in a relationship (relationship status); reporting a 100% heterosexual, mostly heterosexual, bisexual, mostly homosexual, and 100% homosexual orientation (sexual orientation); and identifying as either White, African American or Black, Asian or Asian American, and multiracial (race). Differences between Hispanic and non-Hispanic respondents were examined using *t*-test procedures. All analyses were conducted using SPSS version 26.

## Results

The demographic characteristics for the sample are reported in Table 1. These demographics were similar to those reported by the U.S. Census Bureau for the United States population [24], with some minor differences due to the sampling procedure and exclusion of participants. For example, the age of the survey respondents (from 18 to 70 years old; M = 44.3 years, SD = 16.0 years) is slightly younger than the U.S. census reports as we limited our survey to respondents age 70 years old and younger. However, the proportions of individual age groups based on 5–10 year intervals are consistent with the U.S. population. Almost half of the sample (48.6%) was married, with 34.6% having never been married, and 24.3% not currently dating. While estimates of sexual orientation are not provided by the U.S. Census Bureau, our sample characteristics were similar to those reported by several large studies assessing sexual minority groups [25, 26]. The majority (82.4%) of the sample reported being 100% heterosexual, 14.1% endorsed non-monosexuality (6.9% mostly heterosexual; 1.2% mostly homosexual; 6.0% bisexual), and only 3.0% rated their sexual orientation as 100% homosexual. The small remainder of the sample reported their sexual orientation as questioning (0.2%) or asexual (0.4%). Ethnicity (Hispanic, 17.1% sample vs 17.4% US general population) and racial representation was similar to the US census (67.6% vs 61.9% non-Hispanic White; 11.2% vs 12.3% Black; 4.8% vs 5.3% Asian). The mean educational level was higher than the US general population due to lower rates of people with a high school diploma or less (30.4% sample vs 41.0% US general population). However, the distribution of college degree holders was similar to those of the US general population (Associate's degree, 9.1% sample vs 8.0% US population; Bachelor's degree, 20.7% vs 19.0%; Graduate degree, 11.7% vs 11.0%). Mean household income was lower than the US general population with lower incomes (< $50,000) overrepresented (45.0% vs 40.0%), middle incomes ($50,000 to $75,000) comparably represented (20.7% vs 19.0%), and higher incomes (> $75,000) underrepresented (34.3% vs 41.0%).

### Prevalence of oral, vaginal, and anal sex

Descriptive statistics for the prevalence rates of oral, vaginal, and anal sex are presented in Table 2. Fig 1 graphically depicts the age and sex-related trends in the prevalence of these sexual behaviors. Descriptive tables and figures include 1987 respondents organized by gender (953 men, 1015 women, and 19 that reported non-binary gender), and all analyses were conducted using biological sex (950 males, 1037 females) as the predictor variable. Parameter estimates for each regression model are reported in the Supplemental tables (see https://osf.io/g8phr/?view_only=1a0c0ff0697a406081e25b0f608795a6 for all supplemental tables).

The lifetime prevalence rates of engaging in oral (82.2%) and vaginal (87.0%) sex were higher than that of anal sex (37.9%) (Fig 1). We detected a significant main effect for Age for lifetime vaginal and anal sex, though in opposite directions. Specifically, older age was associated with higher rates of lifetime vaginal sex but lower rates of lifetime anal sex. We also detected significant main effects of $Age^2$ for lifetime oral, vaginal, and anal sex indicating that rates increased from the youngest ages (18–24 years old) and then leveled off in young and middle adulthood (ages 30–59 years old). Finally, we detected a significant Age x Sex interaction for lifetime anal sex, wherein rates were highest from ages 30–39 for females and ages 40–49 for males, with a greater decline at older ages for females relative to males.

For both the past 12 months and past 30 days, about half the sample reported engaging in oral (55.4% and 43.9%, respectively) and vaginal (62.5% and 53.4%, respectively) sex, while rates of anal sex were much lower (15.5% and 11.9%, respectively). Engagement in each sexual behavior over the past 12 months increased from ages 18–24 years old to peak levels at ages 25–39 and then decreased in the older age ranges. Females engaged in anal sex at lower rates

**Table 2. Prevalence rates of oral, vaginal, and anal sex for lifetime, past 12 months, and past 30 days (N = 1,987; men n = 953, women n = 1015).**

| Ages (Years) | Total (%) | Men (%) | | | | | | | Women (%) | | | | | | |
|---|---|---|---|---|---|---|---|---|---|---|---|---|---|---|---|
| | | 18–24 | 25–29 | 30–39 | 40–49 | 50–59 | 60–70 | Total | 18–24 | 25–29 | 30–39 | 40–49 | 50–59 | 60–70 | Total |
| **Oral sex** | | | | | | | | | | | | | | | |
| Lifetime* | 82.2 | 69.4 | 79.4 | 81.8 | 82.6 | 90.0 | 87.4 | **84.1** | 68.6 | 78.5 | 86.9 | 84.2 | 84.7 | 78.5 | **80.7** |
| Past Year*‡‡ | 55.4 | 62.4 | 66.7 | 71.4 | 65.8 | 52.9 | 40.2 | **55.4** | 58.3 | 71.9 | 71.7 | 59.6 | 40.0 | 22.8 | **55.4** |
| Past 30 Days*‡‡ | 43.9 | 50.6 | 58.7 | 63.0 | 52.3 | 43.5 | 27.3 | **44.2** | 50.3 | 54.5 | 57.8 | 44.8 | 31.3 | 16.1 | **43.7** |
| **Vaginal sex** | | | | | | | | | | | | | | | |
| Lifetime* | 87.0 | 68.2 | 87.3 | 85.7 | 86.5 | 91.8 | 93.6 | **88.1** | 68.6 | 80.2 | 92.0 | 92.3 | 93.3 | 91.3 | **86.6** |
| Past Year*‡^ | 62.5 | 61.2 | 71.4 | 75.3 | 72.9 | 58.9 | 50.6 | **62.0** | 60.6 | 74.4 | 78.5 | 69.9 | 52.0 | 36.2 | **63.3** |
| Past 30 Days*‡ | 53.4 | 54.1 | 61.9 | 68.8 | 63.9 | 44.7 | 39.2 | **51.8** | 55.4 | 62.0 | 70.9 | 61.2 | 45.3 | 26.2 | **55.1** |
| **Anal sex** | | | | | | | | | | | | | | | |
| Lifetime‡‡ | 37.9 | 23.5 | 41.3 | 41.6 | 47.7 | 46.5 | 33.4 | **39.0** | 26.9 | 38.8 | 48.1 | 46.4 | 32.7 | 24.2 | **37.2** |
| Past Year*‡† | 15.5 | 18.8 | 33.3 | 29.2 | 26.5 | 12.9 | 4.3 | **16.7** | 18.9 | 19.8 | 20.7 | 14.8 | 6.0 | 3.4 | **14.5** |
| Past 30 Days*‡† | 11.9 | 16.5 | 25.4 | 24.7 | 23.2 | 10.0 | 3.7 | **14.0** | 12.6 | 14.0 | 13.5 | 10.4 | 5.3 | 2.7 | **10.0** |

**Note.** * = main effect of age

‡ = main effect of age$^2$

† = main effect of sex

‡ = significant age x sex interaction term

^ = significant age$^2$ x sex interaction term.

All significant effects $p < .005$.

than males during the past 12 months and past 30 days. Finally, rates of oral and vaginal sex declined with age at a greater rate for females relative to males.

## Frequency of oral, vaginal, and anal sex

Next, we examined age-related trends and sex differences in the frequency of oral, vaginal, and anal sex in the past 12 months and the past 30 days. Descriptive statistics for the frequency of

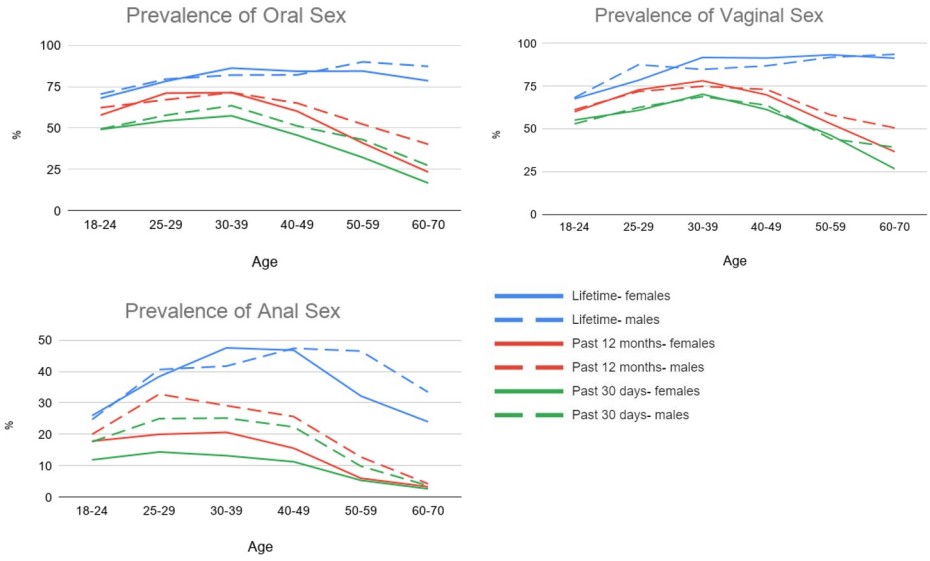

**Fig 1. Age trends in the prevalence rates of oral, vaginal, and anal sex for lifetime, past 12 months, and past 30 days (N = 1,987; men n = 953, women n = 1015).**

**Table 3. Frequency of oral, vaginal, and anal sex in the past 12 months and past 30 days ($N$ = 1,987; men $n$ = 953, women $n$ = 1015).**

| Frequency | Men (%) | | | | | | | | | Women (%) | | | | | | | | | Men vs. Women |
| --- | --- | --- | --- | --- | --- | --- | --- | --- | --- | --- | --- | --- | --- | --- | --- | --- | --- | --- | --- |
| | Mean (SD) | 0 Not at all | 1 <1x month | 2 1x month | 3 >1x month | 4 1x week | 5 >1x week | 6 1x day | 7 >1x day | Mean (SD) | 0 Not at all | 1 <1x month | 2 1x month | 3 >1x month | 4 1x week | 5 >1x week | 6 1x day | 7 >1x day | $d$ |
| **Oral sex** | | | | | | | | | | | | | | | | | | | |
| Past 12 months | 1.6 (1.9) | 44.6 | 15.5 | 8.7 | 14.4 | 6.0 | 9.7 | 1.9 | 1.3 | 1.6 (1.9) | 44.6 | 14.5 | 8.6 | 15.6 | 4.3 | 10.0 | 1.8 | 0.6 | 0.00 |
| Past 30 Days | 1.4 (1.9) | 55.8 | 6.2 | 8.5 | 12.5 | 6.8 | 7.1 | 1.9 | 1.2 | 1.4 (1.9) | 56.3 | 4.8 | 9.2 | 13.6 | 4.7 | 9.1 | 2.0 | 0.4 | 0.00 |
| **Vaginal sex** | | | | | | | | | | | | | | | | | | | |
| Past 12 months | 1.9 (2.0) | 38.0 | 14.0 | 8.2 | 17.2 | 6.6 | 13.2 | 1.4 | 1.5 | 2.2 (2.1) | 36.7 | 10.1 | 7.3 | 17.9 | 6.8 | 16.5 | 3.0 | 1.7 | -0.15 |
| Past 30 Days | 1.8 (2.0) | 48.2 | 5.4 | 8.4 | 15.4 | 7.6 | 12.6 | 0.9 | 1.6 | 2.0 (2.1) | 44.9 | 5.4 | 7.4 | 15.2 | 7.9 | 15.8 | 2.7 | 1.7 | -0.10 |
| **Anal sex** | | | | | | | | | | | | | | | | | | | |
| Past 12 months | 0.5 (1.3) | 83.3 | 4.1 | 3.5 | 3.7 | 2.1 | 2.1 | 0.6 | 0.6 | 0.3 (1.0) | 85.5 | 7.3 | 2.3 | 2.4 | 1.0 | 1.0 | 0.3 | 0.3 | 0.17* |
| Past 30 Days | 0.5 (1.3) | 86.0 | 1.7 | 3.5 | 3.9 | 1.8 | 1.9 | 0.5 | 0.7 | 0.3 (0.9) | 90.0 | 3.7 | 2.4 | 1.4 | 1.1 | 1.0 | 0.4 | 0.1 | 0.18* |

**Note.** Effect sizes of group differences are based on $t$-test procedures.

* $p$ < .005. Cohen's $d$ = X-Y/$\sqrt{((X^2+Y^2)/2)}$. Significant main effects for Age, Age$^2$, Sex, and an Age$^2$ x Sex interaction.

oral, vaginal, and anal sex are reported in Table 3, and Fig 2 provides a graphical depiction of age and sex-related trends in the frequency of these sexual behaviors (parameter estimates from multiple regression models are reported in S2 Table). For the past 12 months, the average frequency of vaginal sex was roughly once a month, between once a month and less than once

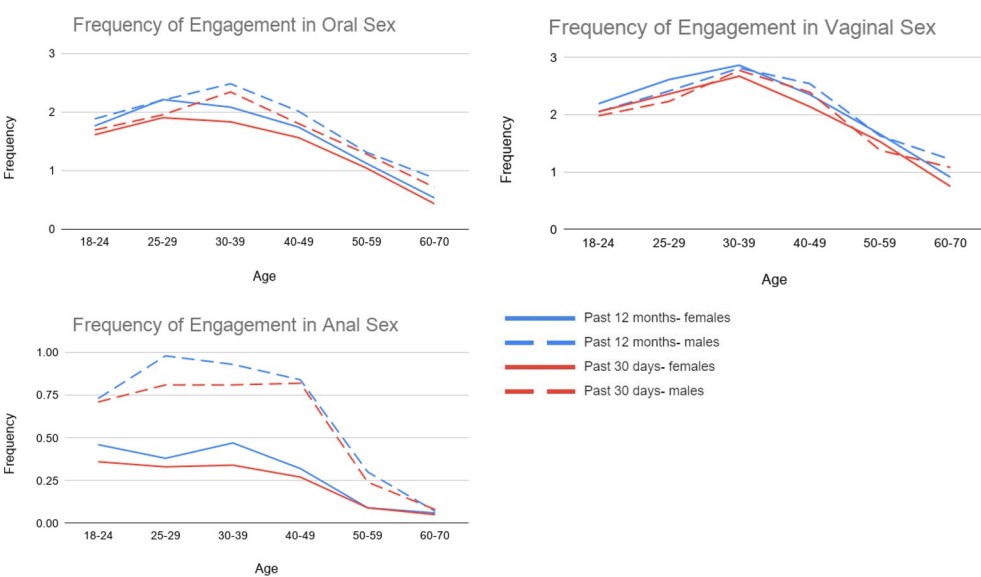

**Fig 2. Age trends in the mean frequency of oral, vaginal, and anal sex in the past 12 months and past 30 days ($N$ = 1,987; men $n$ = 953, women $n$ = 1015).**

a month for oral sex, and less than once a month for anal sex. The frequency of all three sexual behaviors was slightly lower for the past 30 days. Notably, about half of the sample had not engaged in oral sex during the past 12 months or past 30 days (44.6% and 56.1%, respectively). Over one-third of respondents (37.4%) had not engaged in vaginal sex in the past 12 months, and nearly half (46.6%) had not engaged in vaginal sex in the past 30 days. Also, most respondents had not engaged in anal sex during the past 12 months or past 30 days (84.4% and 88.0%, respectively). A small group of males (~3%) and females (4% to 5%) reported engaging in vaginal or oral sex once a day or more in the past 12 months or past 30 days, while an even smaller group reported engaging in anal sex once a day or more (1.2% of men and ~0.5% of women).

The frequency of oral and vaginal sex in the past 12 months and past 30 days increased from ages 18–24 years through age 30–39 years old and then declined after age 40 (after age 30 for the frequency of oral sex in women) with a greater age-related decline for females relative to males. Younger respondents and males engaged in anal sex more frequently during the past 12 months and past 30 days.

## Number of oral, vaginal, and anal sex partners

Next, we examined age-related trends and sex differences in the number of oral, vaginal, and anal sex partners for lifetime and past 12 months. Descriptive statistics are provided in Table 4, and Fig 3 provides a graphical depiction of age-related trends in the number of sexual partners (see S3 Table for parameter estimates from regression models). For the total sample, the mean number of lifetime partners was 8.5 (SD = 15.6) for oral sex, 11.4 (SD = 18.5) for vaginal sex, and 2.1 (SD = 6.5) for anal sex. Notably, a small group of respondents reported an especially high number of sexual partners: 2.4% and 3.9% reported more than 50 lifetime oral and vaginal sex partners, respectively, and 3.6% reported more than 15 lifetime anal sex partners. The number of lifetime oral and vaginal sex partners increased with age until 60 years old for males and 40 years old for females, with a declination thereafter. Males reported significantly more lifetime partners than females for oral (11.9 vs 6.2), vaginal (13.8 vs 9.2), and anal (3.1 vs 1.1) sex (mean $d$ = 0.29). The sex differences for the number of oral and vaginal sex partners increased with age.

The number of sex partners for the past 12 months was lower, but consistent with the pattern observed for the number of lifetime partners for oral, vaginal, and anal sex. Younger respondents reported a greater number of oral and vaginal sex partners, and males reported a greater number of partners for each sexual behavior (mean $d$ = 0.18).

## Age at sexual initiation

Descriptive statistics are provided in Table 5, and Fig 4 provides a graphical depiction of age-related trends in the age of initiation for oral, vaginal, and anal sex (see S4 Table for parameter estimates from regression models). Vaginal sex had the youngest age of initiation (Mean = 18.5 years, SD = 4.7 years), though the mean age of initiation of oral sex was only about one year older (Mean = 19.5 years, SD = 5.9 years) ($d$ = -0.19). Age of initiation of anal sex was much older (Mean = 24.3 year, SD = 8.1 years) than that of vaginal sex ($d$ = 0.87) and oral sex ($d$ = 0.67). There was considerable variability underlying these mean trends. For example, roughly 7% of males and 5% of females reported initiation of oral or vaginal sex by 13 years old, while 4.6% of males and 0.5% of females reported initiation of anal sex by age 13. For initiation at older ages, only 1.4% of males and 0.9% of females reported initiation of oral sex after 40 years old, and less than 1% of males (0.4%) and females (0.7%) reported initiation of vaginal sex after 40 years old. A higher percentage of males (4.6%) and females (2.9%) reported initiation of anal sex after 40 years old. Generally, older participants initiated sexual

**Table 4. Number of oral, vaginal, and anal sex partners for lifetime and past 12 months.**

| | Total | Men | Women | Men vs. Women ($d$) |
|---|---|---|---|---|
| | ($N = 1987$) | ($n = 953$) | ($n = 1015$) | |
| **Lifetime** | | | | |
| **Oral sex** | | | | |
| Mean | 8.5 | 10.9 | 6.1 | 0.31* |
| Median | 3.0 | 4.0 | 3.0 | |
| SD | 15.6 | 18.3 | 11.7 | |
| Range | 0 to 2,000 | 0 to 2,000 | 0 to 1,000 | |
| **Vaginal sex** | | | | |
| Mean | 11.4 | 13.8 | 9.2 | 0.25* |
| Median | 5.0 | 5.0 | 4.0 | |
| SD | 18.5 | 20.8 | 15.5 | |
| Range | 0 to 2,000 | 0 to 2,000 | 0 to 150 | |
| **Anal sex** | | | | |
| Mean | 2.1 | 3.1 | 1.1 | 0.31* |
| Median | 0.0 | 0.0 | 0.0 | |
| SD | 6.5 | 8.4 | 3.4 | |
| Range | 0 to 500 | 0 to 500 | 0 to 50 | |
| **Past 12 months** | | | | |
| **Oral sex** | | | | |
| Mean | 1.1 | 1.3 | 0.9 | 0.19* |
| Median | 1.0 | 1.0 | 1.0 | |
| SD | 2.4 | 2.9 | 1.6 | |
| Range | 0 to 75 | 0 to 75 | 0 to 35 | |
| **Vaginal sex** | | | | |
| Mean | 1.2 | 1.4 | 1.1 | 0.15* |
| Median | 1.0 | 1.0 | 1.0 | |
| SD | 2.4 | 2.8 | 1.9 | |
| Range | 0 to 48 | 0 to 48 | 0 to 30 | |
| **Anal sex** | | | | |
| Mean | 0.5 | 0.7 | 0.3 | 0.19* |
| Median | 0.00 | 0.0 | 0.0 | |
| SD | 2.3 | 2.9 | 1.3 | |
| Range | 0 to 60 | 0 to 60 | 0 to 24 | |

**Note.** Effect sizes of group differences are based on *t*-test procedures.

* $p < .005$. Cohen's $d = X-Y/\sqrt{((X^2+Y^2)/2)}$. For analyses, maximum values were adjusted to the following: 100 for lifetime oral and vaginal sex partners; 45 for lifetime anal sex partners; 20 for past 12 months oral and anal sex partners; 17 for past 12 month vaginal sex partners.

behavior later in life than younger participants. However, this may be partially attributable to some younger participants not having yet initiated certain sexual behaviors, but who will initiate at older ages. Further, females initiated oral and anal sex earlier than males (after adjusting for age and age x sex interactions).

## Additional demographic comparisons

**Relationship status.** Descriptive statistics and effect sizes of group differences in relationship status are provided in Table 6. We detected several significant differences in sexual behaviors based on relationship status. Compared to participants who were not dating (26.6%),

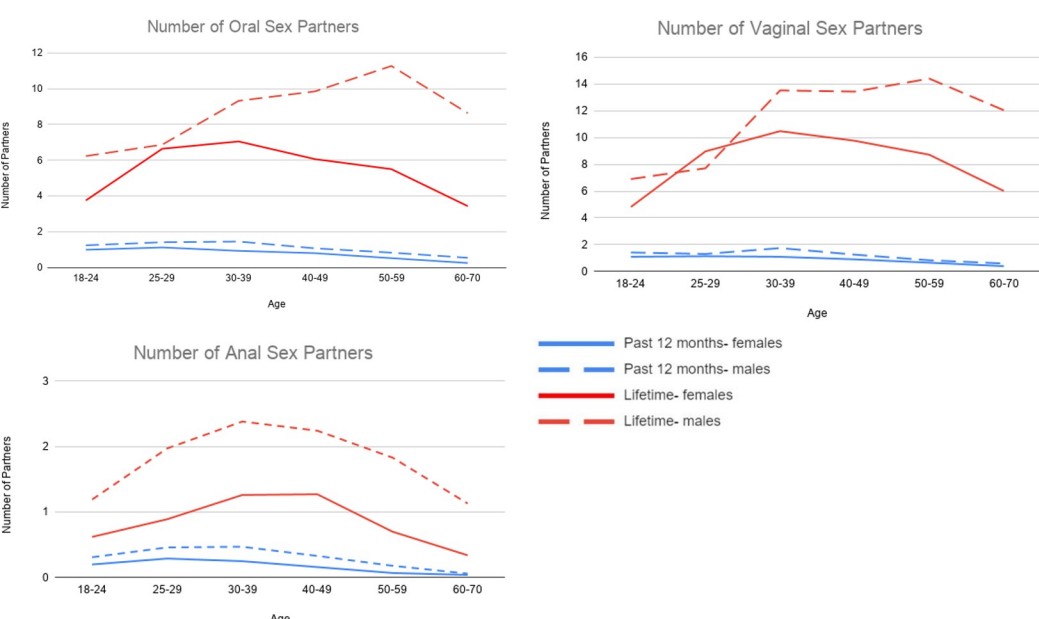

**Fig 3. Age trends in mean number of oral, vaginal, and anal sex partners for lifetime and past 12 months.**

those in a serious relationship (67.7%) reported a higher frequency of oral, vaginal, and anal sex during the past 12 months (mean $d = 0.83$) and the past 30 days (mean $d = 0.76$). This pattern was similar when comparing participants who were not dating to participants who were casually dating (4.6%) during the past 12 months (mean $d = 0.81$) and the past 30 days (mean $d = 0.69$), except for anal sex during the past 30 days. There were no significant differences between respondents who were casually dating and respondents in a serious relationship in the frequency of oral, vaginal, and anal sex in the past 12 months (mean $d = 0.00$) and past 30 days (mean $d = 0.05$).

**Table 5. Age of initiation of oral, vaginal, and anal sex (N = 1,987).**

|  | Total ($n$ = 1,987) Years | Men ($n$ = 953) Years | Women ($n$ = 1,015) Years | Men vs. Women Cohen's $d$ |
|---|---|---|---|---|
| **Oral sex** |  |  |  |  |
| Mean | 19.5 | 19.8 | 19.3 | 0.07 |
| Median | 18.0 | 18.0 | 23.0 |  |
| SD | 5.9 | 6.4 | 5.4 |  |
| Range | 10 to 65 | 10 to 65 | 10 to 60 |  |
| **Vaginal sex** |  |  |  |  |
| Mean | 18.5 | 18.6 | 18.4 | 0.05 |
| Median | 18.0 | 18.0 | 23.0 |  |
| SD | 4.7 | 4.9 | 4.6 |  |
| Range | 10 to 64 | 10 to 64 | 10 to 55 |  |
| **Anal sex** |  |  |  |  |
| Mean | 24.3 | 24.3 | 24.3 | 0.00 |
| Median | 22.0 | 22.0 | 23.0 |  |
| SD | 8.1 | 8.7 | 7.5 |  |
| Range | 10 to 60 | 10 to 60 | 13 to 55 |  |

**Note.** Cohen's $d$ = X-Y/$\sqrt{((X^2+Y^2)/2)}$. Youngest age capped at 10 years old.

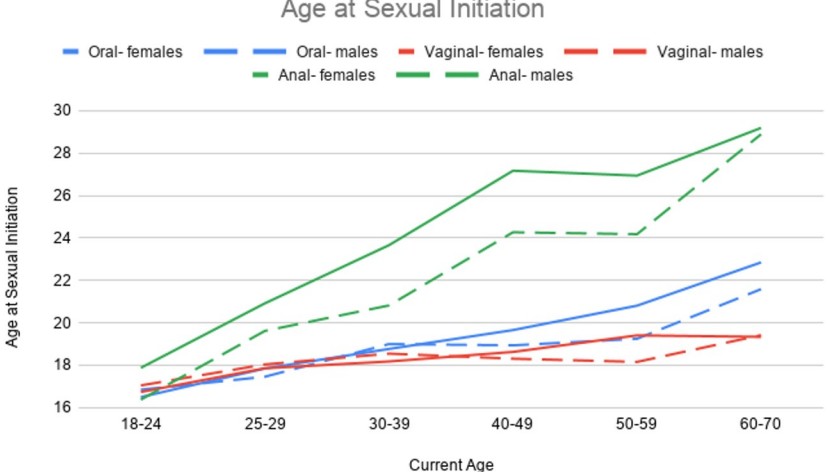

**Fig 4. Age trends in age of initiation in oral, vaginal, and anal sex.**

Compared to participants who were not dating, participants who were casually dating reported a greater number of lifetime partners for oral and vaginal sex (mean $d = 0.40$), and a greater number of partners for oral, vaginal, and anal sex in the past 12 months (mean $d = 0.59$). Participants in a serious relationship reported more lifetime partners for vaginal and anal sex (mean $d = 0.20$) and a greater number of oral, vaginal, and anal sex partners in the past 12 months (mean $d = 0.26$) than those who were not dating. Respondents who were casually dating reported a greater number of lifetime partners for oral sex ($d = 0.30$) and a greater number of oral and vaginal sex partners in the past 12 months (mean $d = 0.50$) than respondents in a serious relationship. Respondents who were casually dating reported an earlier age of initiation for oral, vaginal, and anal sex than those who were not dating (mean $d = -0.47$).

**Sexual orientation.** Descriptive statistics and effect sizes of sexual orientation group differences are provided in Table 7. We found that respondents endorsing 100% heterosexual orientation engaged in vaginal sex more frequently, but in anal sex less frequently, than respondents endorsing 100% homosexual orientation during the past 12 months and past 30 days. Bisexual respondents reported engaging in oral and anal sex more frequently than respondents endorsing 100% heterosexual orientation, and vaginal sex more frequently than respondents endorsing 100% homosexual orientation, for sexual behaviors during the past 12 months and past 30 days.

Respondents endorsing 100% heterosexual orientation reported a greater number of vaginal sex partners, but fewer oral and anal sex partners, than respondents endorsing 100% homosexual orientation. Further, respondents endorsing a bisexual orientation reported a greater number of oral and vaginal sex partners in the past 12 months and a greater number of lifetime oral sex partners than respondents endorsing a 100% heterosexual orientation. Additionally, respondents endorsing a bisexual orientation reported a greater number of vaginal sex partners for lifetime and in the past 12 months but fewer lifetime oral and anal sex partners than respondents endorsing a 100% homosexual orientation.

Respondents endorsing 100% homosexual orientation reported a younger age of initiation of anal sex than respondents endorsing 100% heterosexual orientation. Bisexual respondents reported initiating oral and anal sex at younger ages than participants endorsing 100% heterosexual orientation.

**Table 6. Descriptive statistics and comparisons across relationship status for frequency, number of partners, and age of initiation of oral, vaginal, and anal sex.**

| | Not dating | Casually dating | Serious relationship | Casually dating vs. Not dating | Serious relationship vs. Not dating | Serious relationship vs. Casually dating |
|---|---|---|---|---|---|---|
| | (*n* = 528) | (*n* = 92) | (*n* = 1346) | | | |
| | M (SD) | M (SD) | M (SD) | *d* | *d* | *d* |
| **Frequency** | | | | | | |
| **Past Year** | | | | | | |
| Oral sex | 0.6 (1.2) | 2.1 (1.9) | 2.0 (1.9) | 0.94* | 0.88* | -0.05 |
| Vaginal sex | 0.6 (1.3) | 2.2 (1.9) | 2.6 (2.0) | 0.98* | 1.19* | 0.21 |
| Anal sex | 0.1 (0.7) | 0.7 (1.5) | 0.5 (1.2) | 0.51* | 0.41* | -0.15 |
| **Past 30 days** | | | | | | |
| Oral sex | 0.4 (1.1) | 1.8 (2.0) | 1.8 (2.0) | 0.87* | 0.87* | 0.00 |
| Vaginal sex | 0.5 (1.2) | 1.9 (2.0) | 2.4 (2.1) | 0.85* | 1.11* | 0.24 |
| Anal sex | 0.1 (0.7) | 0.5 (1.4) | 0.4 (1.2) | 0.36 | 0.31* | -0.08 |
| **Number of partners** | | | | | | |
| **Lifetime** | | | | | | |
| Oral sex | 7.5 (15.0) | 14.1 (22.1) | 8.5 (15.2) | 0.35* | 0.07 | -0.30* |
| Vaginal sex | 8.9 (15.2) | 17.9 (24.3) | 12.0 (19.2) | 0.44* | 0.18* | -0.27 |
| Anal sex | 1.2 (3.8) | 2.7 (7.9) | 2.4 (7.1) | 0.24 | 0.21* | -0.04 |
| **Past 12 months** | | | | | | |
| Oral sex | 0.6 (1.9) | 2.7 (3.7) | 1.2 (2.3) | 0.71* | 0.28* | -0.49* |
| Vaginal sex | 0.6 (1.8) | 2.8 (3.6) | 1.3 (2.3) | 0.77* | 0.34* | -0.50* |
| Anal sex | 0.2 (1.0) | 1.0 (3.7) | 0.5 (2.4) | 0.30* | 0.16* | -0.16 |
| **Age at initiation** | | | | | | |
| Oral sex | 20.1 (6.2) | 17.5 (3.6) | 19.5 (5.9) | -0.51* | -0.10 | 0.41 |
| Vaginal sex | 19.0 (5.1) | 17.2 (2.8) | 18.5 (4.7) | -0.44 | -0.10 | 0.34 |
| Anal sex | 26.0 (9.0) | 22.3 (6.8) | 24.0 (7.9) | -0.46 | -0.24 | 0.23 |

**Note.** Effect sizes of group differences are based on post hoc tests using Tukey's procedure.

* *p* < .005. Cohen's *d* = X-Y/$\sqrt{((X^2+Y^2)/2)}$. Not dating = Response options "rarely date" and "not dating now". Casually dating = Response options "mostly going out with one person and dating a few others", "dating or seeing more than one person", and "dating or seeing one person casually". In a serious relationship = Response options "married", "living with someone", "engaged", "planning to get engaged, married, or live together", "in a serious relationship", and "have a boyfriend/girlfriend (and only see each other)". 21 responses were excluded due to inconsistency.

Respondents endorsing mostly heterosexual orientation were most similar to respondents who endorsed bisexual orientation for frequency, number of partners, and age of initiation, especially for oral and vaginal sex. Mostly heterosexual respondents also reported engaging in oral sex significantly more frequently in the past 12 months, a greater number of lifetime oral sex partners, and an earlier age of initiation for oral and vaginal sex than participants reporting a 100% heterosexual orientation. Respondents endorsing a mostly heterosexual orientation reported engaging in anal sex slightly less frequently and had an older age of initiation of anal sex than bisexual participants. Participants reporting mostly homosexual orientation reported more frequent vaginal sex, more vaginal sex partners, fewer lifetime oral sex partners, and older age of initiation of anal sex than participants that reported 100% homosexual orientation (all *d's* > |.50|, but not all were *p* < .005).

**Race and ethnicity.** Descriptive statistics and effect sizes of racial and ethnic group differences are provided in Tables 8 and 9, respectively. We failed to detect any significant differences across racial groups in the frequency of oral, vaginal, or anal sex in the past 12 months and the past 30 days. We detected only one significant difference across racial groups for

**Table 7. Descriptive statistics and comparisons across sexual orientations for frequency, number of partners, and age of initiation of oral, vaginal, and anal sex.**

| | 100% Hetero (*n* = 1631) | Mostly hetero (*n* = 136) | Bi-Sexual (*n* = 119) | Mostly homo (*n* = 23) | 100% Homo (*n* = 60) | 100% Hetero vs. Mostly Heterosexual | 100% Heterosexual vs. Bisexual | Bisexual vs. 100% Homosexual | Mostly homo vs. 100% Homosexual | 100% Hetero vs. 100% Homosexual |
|---|---|---|---|---|---|---|---|---|---|---|
| | M (SD) | M (SD) | M (SD) | M (SD) | M (SD) | *d* | *d* | *d* | *d* | *d* |
| | | | | | **Frequency** | | | | | |
| **Past Year** | | | | | | | | | | |
| Oral sex | 1.5 (1.8) | 2.2 (1.8) | 2.2 (2.0) | 1.5 (1.6) | 2.3 (2.1) | -0.37* | -0.38* | -0.04 | -0.41 | -0.41 |
| Vaginal sex | 2.1 (2.0) | 2.6 (2.0) | 2.4 (2.1) | 1.3 (1.6) | 0.3 (0.9) | -0.24 | -0.16 | 1.33* | 0.82 | 1.15* |
| Anal sex | 0.3 (1.1) | 0.6 (1.1) | 0.9 (1.6) | 0.9 (1.5) | 0.9 (1.5) | -0.20 | -0.40* | -0.04 | -0.04 | -0.45* |
| **Past 30 days** | | | | | | | | | | |
| Oral sex | 1.3 (1.9) | 1.9 (1.9) | 2.0 (2.1) | 1.4 (1.7) | 1.9 (2.2) | -0.30 | -0.32* | 0.03 | -0.28 | -0.28 |
| Vaginal sex | 1.9 (2.1) | 2.3 (2.1) | 2.2 (2.1) | 1.1 (1.6) | 0.2 (0.9) | -0.21 | -0.14 | 1.23* | 0.72 | 1.06* |
| Anal sex | 0.3 (1.0) | 0.4 (0.9) | 0.8 (1.7) | 0.7 (1.3) | 0.9 (1.6) | -0.07 | -0.38* | -0.02 | -0.31 | -0.41* |
| | | | | | **Number of Partners** | | | | | |
| **Lifetime** | | | | | | | | | | |
| Oral sex | 7.2 (13.4) | 12.4 (17.3) | 13.2 (21.7) | 12.0 (21.5) | 25.6 (30.1) | -0.34* | -0.34* | -0.47* | -0.52* | -0.79* |
| Vaginal sex | 11.5 (18.5) | 14.2 (18.0) | 13.4 (22.7) | 8.0 (12.9) | 1.6 (3.3) | -0.15 | -0.09 | 0.73* | 0.67 | 0.75* |
| Anal sex | 1.6 (5.4) | 2.8 (7.5) | 3.4 (7.6) | 6.6 (12.9) | 10.3 (14.7) | -0.18 | -0.27 | -0.59* | -0.27 | -0.79* |
| **Past 12 months** | | | | | | | | | | |
| Oral sex | 1.0 (2.1) | 1.7 (2.7) | 2.2 (3.8) | 2.5 (4.2) | 2.0 (3.9) | -0.31 | -0.41* | 0.05 | 0.13 | -0.34 |
| Vaginal sex | 1.2 (2.1) | 1.6 (2.8) | 2.0 (3.5) | 2.2 (4.8) | 0.1 (0.4) | -0.16 | -0.28* | 0.77* | 0.61* | 0.70 |
| Anal sex | 0.4 (2.0) | 0.6 (2.0) | 0.9 (2.7) | 2.9 (6.4) | 1.6 (4.2) | -0.11 | -0.22 | -0.20 | 0.24 | -0.37* |
| | | | | | **Age of Initiation** | | | | | |
| Oral sex | 19.9 (5.9) | 17.9 (5.5) | 17.4 (4.7) | 19.1 (5.4) | 18.0 (5.7) | 0.35* | 0.47* | -0.11 | 0.19 | 0.33 |
| Vaginal sex | 18.7 (4.8) | 17.0 (3.9) | 17.4 (3.5) | 18.0 (4.3) | 20.1 (6.3) | 0.38* | 0.31 | -0.53 | -0.39 | -0.25 |
| Anal sex | 25.1 (8.2) | 23.3 (8.0) | 20.3 (7.2) | 25.5 (9.5) | 20.3 (5.1) | 0.22 | 0.62* | 0.00 | 0.68 | 0.70* |

**Note**. Effect sizes of group differences are based on post hoc tests using Tukey's procedure. * *p* < .005. Cohen's $d = X\text{-}Y/\sqrt{((X^2+Y^2)/2)}$.

the number of partners in the past 12 months, where Black participants reported more vaginal sex partners than White participants. Black and White participants reported an earlier age of initiation of oral and vaginal sex than Asian participants. Although most comparisons were not significantly different, there was a consistent trend of Black participants > White participants > Asian participants for the frequency of oral and vaginal sex in the past 12 months and past 30 days, and the number of lifetime partners for oral and vaginal sex. The trend was reversed for the age of initiation of oral and vaginal sex. The differences for anal sex were smaller, and the trend was different for the age of initiation of anal (Asian participants < Black participants < White participants).

Hispanic ethnicity was associated with a higher frequency of oral, vaginal, and anal sex in the past 12 months (mean *d* = 0.30) and in the past 30 days (mean *d* = 0.30). Hispanic ethnicity was

**Table 8. Descriptive statistics and standardized effect sizes for group comparisons for racial groups.**

| | Frequency | | | | | | | | |
|---|---|---|---|---|---|---|---|---|---|
| | White | Black | Asian | Mixed | Black vs. White | Black vs. Asian | Asian vs. White | Mixed vs. White | Mixed vs. Black |
| | (*n* = 1536) | (*n* = 233) | (*n* = 86) | (*n* = 49) | | | | | |
| | M (SD) | M (SD) | M (SD) | M (SD) | *d* | *d* | *d* | *d* | *d* |
| **Past 12 months** | | | | | | | | | |
| Oral sex | 1.5 (1.8) | 1.9 (2.1) | 1.3 (1.7) | 1.7 (1.7) | 0.20 | 0.31 | -0.11 | 0.11 | -0.10 |
| Vaginal sex | 2.0 (2.0) | 2.3 (2.1) | 1.8 (2.0) | 2.1 (2.1) | 0.15 | 0.24 | -0.10 | 0.05 | -0.10 |
| Anal sex | 0.4 (1.1) | 0.4 (1.2) | 0.4 (1.1) | 0.4 (0.8) | 0.00 | 0.17 | -0.18 | 0.00 | 0.00 |
| **Past 30 days** | | | | | | | | | |
| Oral sex | 1.4 (1.9) | 1.7 (2.1) | 1.1 (1.7) | 1.7 (2.0) | 0.15 | 0.31 | -0.17 | 0.15 | 0.00 |
| Vaginal sex | 1.8 (2.0) | 2.1 (2.2) | 1.7 (2.1) | 1.9 (2.2) | 0.14 | 0.19 | -0.05 | 0.05 | -0.09 |
| Anal sex | 0.3 (1.1) | 0.3 (1.0) | 0.3 (1.0) | 0.4 (1.0) | 0.00 | 0.00 | 0.00 | 0.10 | 0.10 |
| | Number of Partners | | | | | | | | |
| **Lifetime** | | | | | | | | | |
| Oral sex | 8.5 (15.4) | 9.8 (19.0) | 6.5 (15.2) | 8.9 (11.0) | 0.08 | 0.19 | -0.13 | 0.03 | -0.06 |
| Vaginal sex | 11.5 (18.4) | 14.4 (20.9) | 7.4 (15.6) | 10.4 (19.4) | 0.15 | 0.38 | -0.24 | -0.06 | -0.20 |
| Anal sex | 2.2 (6.6) | 1.7 (6.0) | 2.4 (8.6) | 1.3 (2.3) | -0.08 | -0.09 | 0.03 | -0.18 | -0.09 |
| **Past 12 months** | | | | | | | | | |
| Oral sex | 1.0 (2.2) | 1.5 (2.7) | 1.0 (2.1) | 1.4 (3.0) | 0.20 | 0.21 | 0.00 | 0.15 | -0.04 |
| Vaginal sex | 1.1 (2.1) | 1.7 (3.0) | 1.6 (3.5) | 1.6 (3.4) | 0.23* | 0.03 | 0.17 | 0.18 | -0.03 |
| Anal sex | 0.5 (2.2) | 0.6 (2.7) | 0.6 (2.5) | 0.3 (0.6) | 0.04 | 0.00 | 0.04 | -0.12 | -0.15 |
| | Age at Initiation | | | | | | | | |
| Oral sex | 19.5 (5.8) | 19.0 (5.6) | 22.2 (8.5) | 18.3 (5.2) | -0.09 | -0.44* | 0.37* | -0.22 | -0.13 |
| Vaginal sex | 18.4 (4.4) | 18.1 (6.0) | 21.3 (4.8) | 18.0 (7.5) | -0.06 | -0.59* | 0.63* | -0.07 | -0.01 |
| Anal sex | 24.6 (8.4) | 23.2 (6.7) | 22.5 (7.1) | 22.6 (7.6) | -0.18 | 0.10 | -0.27 | -0.25 | -0.08 |

**Note**. Effect sizes of group differences are based on post hoc tests using Tukey's procedure.

* *p* < .005. Cohen's *d* = X-Y/$\sqrt{((X^2+Y^2)/2)}$. 85 responses were excluded due to incompletion.

also associated with a slightly lower number of lifetime vaginal sex partners, a slightly greater number of anal sex partners during the past 12 months, and an earlier age of initiation of anal sex.

## Discussion

We sought to evaluate the representativeness of the SIPS sample—a national survey conducted to better understand relationships among sexual behavior, internet use, and psychological adjustment—by examining whether the rates and demographic trends in sexual behavior replicated prior estimates from similar national surveys. Sample estimates of the prevalence, frequency, number of partners, and age of initiation in oral, vaginal, and anal sex were broadly consistent with prior studies, as were subgroup analyses examining differences in sexual behavior associated with relationship status, sexual orientation, race, and ethnicity.

### Trends associated with age and sex

Results for age and sex trends in the prevalence, frequency, number of partners, and age of initiation were consistent with and expand upon previous nationally representative online samples [12]. We replicated the finding of high rates of lifetime participation in oral and vaginal sex (>80%), and a much lower, but not trivial, rate of anal sex (<40%). As our subgroup

**Table 9. Descriptive statistics and standardized effect sizes for group comparisons for hispanic vs. non-hispanic groups.**

| | Frequency | | |
| --- | --- | --- | --- |
| | Hispanic ($n$ = 340) | Non-Hispanic ($n$ = 1647) | Hispanic vs. Non-Hispanic |
| | M (SD) | M (SD) | ($d$) |
| **Past 12 months** | | | |
| Oral sex | 2.1 (2.1) | 1.5 (1.8) | 0.31* |
| Vaginal sex | 2.5 (2.2) | 1.9 (2.0) | 0.29* |
| Anal sex | 0.7 (1.5) | 0.3 (1.0) | 0.31* |
| **Past 30 days** | | | |
| Oral sex | 2.0 (2.1) | 1.3 (1.8) | 0.36* |
| Vaginal sex | 2.4 (2.2) | 1.8 (2.0) | 0.29* |
| Anal sex | 0.6 (1.5) | 0.3 (1.0) | 0.24* |
| | **Number of partners** | | |
| **Lifetime** | | | |
| Oral sex | 6.8 (12.9) | 8.8 (16.0) | -0.14 |
| Vaginal sex | 8.5 (15.4) | 12.0 (19.0) | -0.20* |
| Anal sex | 2.5 (7.3) | 2.0 (6.3) | 0.07 |
| **Past 12 months** | | | |
| Oral sex | 1.4 (2.8) | 1.1 (2.3) | 0.11 |
| Vaginal sex | 1.5 (2.8) | 1.2 (2.3) | 0.11 |
| Anal sex | 0.8 (3.0) | 0.4 (2.1) | 0.16* |
| | **Age at initiation** | | |
| Oral sex | 19.2 (5.4) | 19.6 (6.0) | -0.07 |
| Vaginal sex | 18.6 (4.8) | 18.5 (4.7) | 0.02 |
| Anal sex | 22.5 (6.2) | 24.7 (8.5) | -0.30* |

**Note.** Effect sizes of group differences are based on t-test procedures.

* $p < .005$. Cohen's $d$ = X-Y/$\sqrt{((X^2+Y^2)/2)}$.

analyses on sexual orientation revealed, this difference is largely attributable to the low proportion of homosexual men in an unselected national sample. This is to be expected in population samples, where rates of non-heterosexual orientation and male-male sexual behavior are relatively low, and heterosexual couples are more likely to choose to engage in oral and vaginal sex. For example, only 17.6% of respondents reported a non-heterosexual orientation in the current sample, and only 34.4% of respondents endorsing 100% heterosexual orientation reported ever having engaged in anal sex, whereas 61.7% of respondents endorsing 100% homosexual orientation reported engaging in anal sex. The lower levels of anal sex among heterosexual participants may be explained by the stigma associated with anal sex, for example, beliefs that anal sex is immoral or "dirty," along with historical restrictions on activities such as sodomy and same-sex marriage [27, 28].

We also replicated prior findings that the frequency of these sexual behaviors was highest in middle adulthood [12, 29]. The associations between frequency of sex and age may be at least partially attributable to relationship status. That is, people in serious relationships tend to have more frequent sex than single people, and people tend to form more serious relationships in young and middle adulthood. For example, the average age of first marriage in the United States was 29 years old in 2020 [30]. The frequency of sex may also be influenced by fertility and intention to birth children, a process that must occur before menopause (typically, before ages 45–55 years old) [31, 32].

We also replicated prior findings that men reported a greater number of sexual partners than women throughout their lifetime [10, 12, 20]. Men's report of the number of sexual partners also exhibited higher variance than women's report of the number of sexual partners. This was primarily due to a small proportion of men that reported an especially high number of sexual partners, increasing both the mean and variance of the number of sexual partners relative to women [33]. We also identified a considerably lower median value for the number of sexual partners compared to the mean value across gender, further illustrating the influence of the small proportion of respondents reporting a very high number of partners, with the effect being stronger in men compared to women. Further, in a study addressing why estimates of sexual partners are often higher in men than women, Mitchell et al. [34] found that men tend to estimate their number of sexual partners (often rounding up to numbers that end in 0 or 5) whereas women tend to count them, a reporting error which may have affected the point estimates in our sample.

We also found that anal sex had a much later age of initiation compared to oral and vaginal sex [18]. This is consistent with the lower prevalence rate of anal sex relative vaginal and oral sex. That is, anal sex is a less common and more sexually advanced behavior and may require greater preparation compared to oral and vaginal sex [35]. This may pose a barrier to populations with limited access to certain sexual wellness resources (i.e., lubrication).

## Trends associated with relationship status, sexual orientation, race, and ethnicity

We also found that numerous aspects of sexual behavior varied by subgroups of relationship status, sexual orientation, race, and ethnicity. As oral, vaginal, or anal sex are all partnered behaviors, it follows that participating in ongoing romantic and sexual relationships facilitates access to potential sex partners, access that is more limited or at least not as readily available to people not participating in such relationships. This finding is in line with prior research, confirming that while single people may have greater opportunities for casual sex with a greater number of partners, people in a relationship have more frequent sex [14–16]. Interestingly, people that reported being in a casual relationship(s) reported a higher frequency of sex and more sexual partners, though they constituted a small proportion of the sample.

Consistent with prior findings regarding sexual orientation, heterosexual and bisexual men reported higher rates of vaginal sex than homosexual men. This finding is easily understood via anatomical constraints, that is, vaginal sex is not possible in cisgender male-male partnerships. Further, homosexual males reported higher rates of anal sex than heterosexual and bisexual males. As discussed previously, the stigma surrounding anal sex may account for lower rates of anal sex in non-homosexual subgroups. We also found that bisexual respondents participated in a wide variety of sexual behavior and reported a particularly high frequency of sex and earlier age of sexual initiation compared to other sexual orientation groups. Similar trends were also observed for respondents that endorsed a mostly heterosexual or mostly homosexual orientation. Some prior work suggests that sexual minority groups have more liberal attitudes about sex compared to heterosexual people, that is, a greater acceptance of recreational sex, potentially contributing to an earlier age of sexual initiation and a higher frequency of sex [36, 37].

Results for racial differences in sexual behaviors were less consistent with prior findings. We did detect a significant difference for earlier ages of initiation for oral and vaginal sex for White and Black participants relative to Asian participants, and that Black participants reported more lifetime vaginal sex partners than White participants. Although the effects were small (Cohen's $d < .30$), we did observe a *trend* for Black participants to report a higher

frequency, more partners, and an earlier age of initiation for oral and vaginal sex than White and Asian participants as reported in prior studies [14, 15, 21–23]. The lack of significant differences could be due to a different design than most prior studies of general sexual behavior (i.e., our use of internet-based recruitment and survey methods assessing oral, vaginal, and anal sex separately). While Herbenick and colleagues [12] examined oral, vaginal, and anal sex trends in an online-based survey, they did not report on racial differences, which would have provided a useful comparison for the current study and allowed us to explore if racial differences in sexual behavior are smaller or less robust relative to factors such as relationship status and sexual orientation, or if racial differences in sexual behavior in the US are narrowing over time. Additionally, research is limited as to *why* these racial differences in normative sexual behavior may be present. It will be useful for future studies to expand analyses beyond simple group differences, and to test if additional covariates (e.g., cultural attitudes about sex and relationships) can account for any racial differences in these sexual behaviors.

For ethnicity, we also found that Hispanic respondents reported a higher frequency of oral, vaginal, and anal sex than non-Hispanic respondents, as well as slightly fewer lifetime vaginal sex partners and an earlier age of initiation of anal sex. To our knowledge, all prior studies explicitly examining ethnic differences in sexual behavior did so using the umbrella term "sexual intercourse" [14, 15]. We assessed sexual behavior in greater detail, specifying between oral, vaginal, and anal sex, expanding upon prior examinations of ethnic differences in sexual behavior. It will be important to replicate and dismantle such differences in future research.

## Limitations

While the online survey design provided for a highly efficient mode of data of collection for a national sample, several limitations should be noted. Although the Qualtrics XM platform ensures a sample that mirrors the US general population for the quota variables, all participating respondents had either internet or cell phone access. Though the large majority of the U.S. population does have internet access, there remains a small percentage who do not. Also, while Qualtrics guarantees specific quotas, it does not guarantee representation on other non-quota variables. Furthermore, due to time constraints, we focused on assessing oral, vaginal, and anal sex. Other surveys have asked about a wide range of sexual behaviors including solo behaviors, participation in group sex, use of sex toys, and various kink behaviors. While we did not include that breadth, we were able to delve into greater detail of oral, vaginal, and anal sex than some prior reports.

Additionally, multivariate logistic and linear regression models were used to explore the effects of Age, $Age^2$, Sex, and their interactions on oral, vaginal, and anal sex. We did not, however, perform these analyses within demographic subgroups. While one-way ANOVAs were useful in testing for mean differences across subgroups, we did not test for interactions among age, sex, and the other demographic variables. Given we identified numerous group differences in sexual behavior, this will be an important goal for future research, though some subgroup analyses may require larger sample sizes than the SIPS to provide adequate power to detect differences (e.g., non-heterosexual orientation subgroups). There are also other demographic variables that we did not examine (e.g., income, education), as well as other behaviors (e.g., internet use, substance use) and psychological characteristics (e.g., personality) that are associated with sexual behavior, some of which we intend to examine in future reports. Subgroups could also be further dissected in large samples, e.g., sexual attraction as a distinct facet of sexual orientation [38]. There are other demographic variables that we did not examine (e.g., income, education), as well as other behaviors (e.g., internet use, pornography

consumption, substance use) and psychological characteristics (e.g., personality) that may be associated with sexual behavior, some of which we intend to examine in future reports [39, 40].

Overall, the results are consistent with most prior findings regarding patterns of oral, vaginal, and anal sex in the United States, and help establish the representativeness of the SIPS sample. The evidence for its representativeness provides a basis on which future investigations can examine and make valid inferences regarding associations among sexual behavior, technology, and adjustment. Future reports will analyze additional correlates of sexual behavior, including social media and dating app use, substance use, mental health, personality, and interpersonal relationship traits to explore how modern technology usage impacts the expression of sexual and romantic behavior.

## Supporting information

**S1 Table. Effects of age and biological sex on prevalence rates of oral, vaginal and anal sex.** (DOCX)

**S2 Table. Effects of age and biological sex on frequency of oral, vaginal, and anal sex in the past 12 months and past 30 days ($N$ = 1,987; men $n$ = 953, women $n$ = 1015).** (DOCX)

**S3 Table. Effects of age and biological sex on number of oral, vaginal, and anal sex partners for lifetime and the past 12 months ($N$ = 1,987; men $n$ = 953, women $n$ = 1015).** (DOCX)

**S4 Table. Effects of age and biological sex on age of initiation of oral (men $n$ = 801, women $n$ = 819), vaginal (men $n$ = 840, women $n$ = 879), and anal sex partners (men $n$ = 372, women $n$ = 378).** (DOCX)

## Author Contributions

**Conceptualization:** Carter Sherman, Mary M. Heitzeg, Brian M. Hicks.

**Data curation:** Hannah Roberts.

**Formal analysis:** Angus Clark.

**Funding acquisition:** Mary M. Heitzeg, Brian M. Hicks.

**Investigation:** Hannah Roberts, Angus Clark, Brian M. Hicks.

**Methodology:** Angus Clark, Brian M. Hicks.

**Project administration:** Carter Sherman, Brian M. Hicks.

**Resources:** Brian M. Hicks.

**Software:** Angus Clark.

**Supervision:** Angus Clark, Carter Sherman, Brian M. Hicks.

**Validation:** Angus Clark, Brian M. Hicks.

**Writing – original draft:** Hannah Roberts, Brian M. Hicks.

**Writing – review & editing:** Hannah Roberts, Angus Clark, Carter Sherman, Mary M. Heitzeg, Brian M. Hicks.

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
