## [Decision Letter · Decision Letter 0]

9 Jun 2021

PONE-D-21-09417

Age, Sex, and Other Demographic Trends in Sexual Behavior in the United States: Initial Findings of the Sexual Behaviors, Internet Use, and Psychological Adjustment Survey

PLOS ONE

Dear Dr. Roberts,

Thank you for submitting your manuscript to PLOS ONE. After careful consideration, we feel that it has merit but does not fully meet PLOS ONE’s publication criteria as it currently stands. Therefore, we invite you to submit a revised version of the manuscript that addresses the points raised during the review process.

There is a need to address some of the literature that was not referenced in the previous version

We look forward to receiving your revised manuscript.

Kind regards,

Andrew R. Dalby, PhD

Academic Editor

PLOS ONE

Journal Requirements:

2. We note you have included tables to which you do not refer in the text of your manuscript. Please ensure that you refer to Tables 6-9 in your text; if accepted, production will need this reference to link the reader to the Tables.

Reviewers' comments:

Reviewer's Responses to Questions

**Comments to the Author**

1. Is the manuscript technically sound, and do the data support the conclusions?

Reviewer #1: Yes

2. Has the statistical analysis been performed appropriately and rigorously? 

Reviewer #1: Yes

3. Have the authors made all data underlying the findings in their manuscript fully available?

Reviewer #1: No

4. Is the manuscript presented in an intelligible fashion and written in standard English?

Reviewer #1: Yes

5. Review Comments to the Author

Reviewer #1: The article, “Age, Sex, and Other Demographic Trends in Sexual Behavior in the United States: Initial 7 Findings of the Sexual Behaviors, Internet Use, and Psychological Adjustment Survey” presents comparisons of sexual behaviors by demographic identities. Although the methodology appears strong, I have some reservations about the article’s contributions to the literature. Overall, the purpose of ensuring that this sample is representative is not a compelling reason to publish an article – though I’m sure it would be helpful for the team to reference in subsequent publications. The manuscript overall is missing a rationale for why this study is needed – based on this sample being unique (which seems antithetical to the authors’ goals) or collecting new information in a new way, etc. There appears to be a slim rationale that Herbenick et al. did not compare racial groups, but this isn’t articulated until the discussion.

General:

Please perform a careful grammatical review of the paper. I have included a few examples below, but did not have the ability to completely review for this.

P3L64: “social media is continues to rise”

Table 1 footnote “ands”

Please also review for consistency in citation – such as putting the numeric citation in parentheses (example: P9L195)

Abstract:

The abstract results section appears to not deliver on the promise of focusing on app usage. The authors report only sexual behaviors based on relationship, race/ethnicity, and sexual identity.

Introduction

I would encourage the authors to include more of the publications based on the National Survey of Sexual Health and Behavior – they have a few from the 2010 dataset, but there are further comparisons that I believe are missed.

Levine, E. C., Herbenick, D., Martinez, O., Fu, T. C., & Dodge, B. (2018). Open relationships, nonconsensual nonmonogamy, and monogamy among US adults: Findings from the 2012 National Survey of Sexual Health and Behavior. Archives of sexual behavior, 47(5), 1439-1450.

Herbenick, D., Fu, T. C., Wright, P., Paul, B., Gradus, R., Bauer, J., & Jones, R. (2020). Diverse Sexual Behaviors and Pornography Use: Findings From a Nationally Representative Probability Survey of Americans Aged 18 to 60 Years. The journal of sexual medicine, 17(4), 623-633.

Fu, T. C., Herbenick, D., Dodge, B., Owens, C., Sanders, S. A., Reece, M., & Fortenberry, J. D. (2019). Relationships among sexual identity, sexual attraction, and sexual behavior: Results from a nationally representative probability sample of adults in the United States. Archives of sexual behavior, 48(5), 1483-1493.

Dodge, B., Herbenick, D., Fu, T. C. J., Schick, V., Reece, M., Sanders, S., & Fortenberry, J. D. (2016). Sexual behaviors of US men by self-identified sexual orientation: Results from the 2012 National Survey of Sexual Health and Behavior. The journal of sexual medicine, 13(4), 637-649.

Methods

Participants reported their biological sex, not their gender identity. I would recommend the use of male/female to indicate sex rather than man/woman which refers to gender.

In terms of reporting relationship status, what about participants that were non-monogamous? Were they allowed to select more than one response option, and if so, how were their responses collapsed into the three categories?

Was oral, vaginal, and anal sex defined for participants? For instance, was digital penetration considered sex?

P9L188: Unclear what exactly is meant here. Should one be ‘mostly homosexual’? “reporting a 100% heterosexual, mostly heterosexual, bisexual, mostly heterosexual, …”

Results

P9L202: Rather than using the phrase “sexual fluid” I would recommend non-monosexual. Sexual fluidity connotes change, whereas many participants who identify as bisexual may not report any shifts – but rather a consistent attraction to more than one gender.

For parity, I would suggest that the authors also reference national population estimates for sexual identity as they do with other demographic categories. The following is a strong source for those estimates: https://williamsinstitute.law.ucla.edu/publications/how-many-people-lgbt/

On page 9, suddenly there are nonbinary participants that were not included in the description of demographic measures – the authors only discuss asking about biological sex.

P17L296: “The number of lifetime partners increased with age and then reached a plateau in 296 older men” – can the authors include a parenthetical here that defines what older men means?

Discussion

P30L488: Please rephrase the following to indicate “intention to BIRTH children”. People can have children (including trans men) in all types of ways that are not restricted by menopause, including surrogacy or adoption. “The frequency of sex may also be influenced by fertility and intention to have children, a process that must occur for women before menopause (typically, before ages 45-55 years old).”

I would recommend that the authors pull in Savin-Williams work on “mostly heterosexual” men. Austin has done some work related to “mostly heterosexual” women as well.

6. PLOS authors have the option to publish the peer review history of their article (what does this mean?). If published, this will include your full peer review and any attached files.

Reviewer #1: No

---

## [Author Response · Author response to Decision Letter 0]

12 Jul 2021

Reviewer Overview: The article, “Age, Sex, and Other Demographic Trends in Sexual Behavior in the United States: Initial Findings of the Sexual Behaviors, Internet Use, and Psychological Adjustment Survey” presents comparisons of sexual behaviors by demographic identities. Although the methodology appears strong, I have some reservations about the article’s contributions to the literature. Overall, the purpose of ensuring that this sample is representative is not a compelling reason to publish an article – though I’m sure it would be helpful for the team to reference in subsequent publications. The manuscript overall is missing a rationale for why this study is needed – based on this sample being unique (which seems antithetical to the authors’ goals) or collecting new information in a new way, etc. There appears to be a slim rationale that Herbenick et al. did not compare racial groups, but this isn’t articulated until the discussion.

Author Response: We appreciate your time spent reviewing our manuscript. We agree that the rationale was vague and have since altered multiple sentences in the Introduction to resolve this issue. We posit that replication, especially of research which utilized relatively modern sampling methods, provides considerable contributions to the literature equal in value to novelty. Also, we provide more detail than some prior studies including information on frequency, number of partners, and age of initiation for sexual behaviors as well as comparisons across relationship status, sexual orientation, race, and ethnicity. Therefore, we have revised the text to highlight the importance of replication:

“We also sought to replicate and expand upon recent surveys that reported nationally representative demographic trends in sexual behavior to validate the representativeness of the SIPS sample and bolster the reliability and validity of both past estimates of sexual behavior. It is especially critical to replicate findings which utilize relatively novel, modern sampling methods, i.e., internet-based national surveys. Thus, as participation in internet-based social networking continues to rise, it is critical to understand if and how these social trends are impacting in-person social relationships, sexual behaviors, and well-being more broadly.”

Reviewer Comment 1: General: Please perform a careful grammatical review of the paper. I have included a few examples below, but did not have the ability to completely review for this.

P3L64: “social media is continues to rise”

Table 1 footnote “ands”

Please also review for consistency in citation – such as putting the numeric citation in parentheses (example: P9L195)

Author Response 1: We have carefully reviewed the manuscript and tables for grammatical and citation errors. We are certain that all documents meet expectations.

Reviewer Comment 2: Abstract: The abstract results section appears to not deliver on the promise of focusing on app usage. The authors report only sexual behaviors based on relationship, race/ethnicity, and sexual identity.

Author Response 2: We thank you for this comment. We have replaced app usage with the broader term of “internet”. See excerpt below:

“It remains unclear how the seemingly ubiquitous use of the internet impacts user’s off-line personal relationships, particularly those that are romantic or sexual.”

Reviewer Comment 3: Introduction: I would encourage the authors to include more of the publications based on the National Survey of Sexual Health and Behavior – they have a few from the 2010 dataset, but there are further comparisons that I believe are missed.

Levine, E. C., Herbenick, D., Martinez, O., Fu, T. C., & Dodge, B. (2018). Open relationships, nonconsensual nonmonogamy, and monogamy among US adults: Findings from the 2012 National Survey of Sexual Health and Behavior. Archives of sexual behavior, 47(5), 1439-1450.

Herbenick, D., Fu, T. C., Wright, P., Paul, B., Gradus, R., Bauer, J., & Jones, R. (2020). Diverse Sexual Behaviors and Pornography Use: Findings From a Nationally Representative Probability Survey of Americans Aged 18 to 60 Years. The journal of sexual medicine, 17(4), 623-633.

Fu, T. C., Herbenick, D., Dodge, B., Owens, C., Sanders, S. A., Reece, M., & Fortenberry, J. D. (2019). Relationships among sexual identity, sexual attraction, and sexual behavior: Results from a nationally representative probability sample of adults in the United States. Archives of sexual behavior, 48(5), 1483-1493.

Dodge, B., Herbenick, D., Fu, T. C. J., Schick, V., Reece, M., Sanders, S., & Fortenberry, J. D. (2016). Sexual behaviors of US men by self-identified sexual orientation: Results from the 2012 National Survey of Sexual Health and Behavior. The journal of sexual medicine, 13(4), 637-649.

Author Response 3: Thank you for this wealth of resources. We have reviewed and utilized all manuscripts. We have added sentences to P5L96, L100 of the Introduction and page P34L580, L579 of the Discussion. See excerpts below:

“For example, 4-8% of individuals report participation in non-monogamous relationships, allowing for the possibility of multiple sexual and romantic partners (Levine, Herbenick, Martinez, Fu & Dodge, 2018).”

“Further, homosexual and bisexual men reported lower rates of vaginal sex, but higher rates of insertive and receptive anal sex compared to heterosexual men. (Dodge et al., 2016).”

“There are other demographic variables that we did not examine (e.g., income, education), as well as other behaviors (e.g., internet use, pornography consumption, substance use) and psychological characteristics (e.g., personality) that may be associated with sexual behavior, some of which we intend to examine in future reports (Herbenick et al., 2020; Moussa Rogers, Kelley & McKinney, 2021).”

“Subgroups could also be further dissected in large samples, e.g., sexual attraction as a distinct facet of sexual orientation (Fu et al., 2018).”

Reviewer Comment 4: Methods: Participants reported their biological sex, not their gender identity. I would recommend the use of male/female to indicate sex rather than man/woman which refers to gender.

Author Response 4: We have made this alteration where applicable.

Reviewer Comment 5: In terms of reporting relationship status, what about participants that were non-monogamous? Were they allowed to select more than one response option, and if so, how were their responses collapsed into the three categories?

Author Response 5: Participants received the question “Which best describes your relationship status now?” with the following 11 response options: (1) Married, (2) Living with someone, (3) Engaged, (4) Planning to get engaged, married, or live together, (5) In a serious relationship with one person, (6) Have a boyfriend or girlfriend (and only see each other), (7) Mostly going out with one person and dating a few others, (8) Dating or seeing more than one person, (9) Dating or seeing one person casually, (10) Rarely date, (11) Not dating now”. Respondents were able to select multiple response options if applicable. 

The category “serious relationship” included responses 1-6. The category “casual relationship” included responses 7, 8, and 9. Finally, the category “Not dating or not in a relationship” included responses 10 and 11. We have clarified this in the Measures section. See the excerpt below:

“Although we provided 11 possible response options, these options were collapsed into three categories for analysis. Respondents were able to select multiple response options if applicable. Not dating or not in a relationship included the responses: rarely date and not dating now. Being in a casual relationship included the responses: mostly going out with one person and dating a few others, dating or seeing more than one person, and dating or seeing one person casually. Being in a serious relationship included the responses: married; living with someone; engaged; planning to get engaged, married, or live together; in a serious relationship; and have a monogamous boyfriend/girlfriend.”

Reviewer Comment 6: Was oral, vaginal, and anal sex defined for participants? For instance, was digital penetration considered sex?

Author Response 6: Oral sex was defined as when “a person puts their mouth on another person’s sex organs” and vaginal sex was defined as “inserting the penis into the vagina”. Anal sex was described as when “a man inserts his penis into his partner’s anus or asshole”. See excerpt below:

“First, we asked respondents to report if they had ever engaged in oral, vaginal, and anal sex (yes/no) to assess prevalence rates for each behavior. Oral sex was defined as when “a person puts their mouth on another person’s sex organs” and vaginal sex was defined as “inserting the penis into the vagina”. Anal sex was described as when “a man inserts his penis into his partner’s anus or asshole”.”

Reviewer Comment 7: P9L188: Unclear what exactly is meant here. Should one be ‘mostly homosexual’? “reporting a 100% heterosexual, mostly heterosexual, bisexual, mostly heterosexual, …”

Author Response 7: We apologize for this error. It has been fixed. See excerpt below:

“...reporting a 100% heterosexual, mostly heterosexual, bisexual, mostly homosexual, and 100% homosexual orientation (sexual orientation)...”

Reviewer Comment 8: Results: P9L202: Rather than using the phrase “sexual fluid” I would recommend non-monosexual. Sexual fluidity connotes change, whereas many participants who identify as bisexual may not report any shifts – but rather a consistent attraction to more than one gender.

Author Response 8: This change has been made. See excerpt below:

“The majority (82.4%) of the sample reported being 100% heterosexual, 14.1% endorsed non-monosexuality (6.9% mostly heterosexual; 1.2% mostly homosexual; 6.0% bisexual), and only 3.0% rated their sexual orientation as 100% homosexual.”

Reviewer Comment 9: For parity, I would suggest that the authors also reference national population estimates for sexual identity as they do with other demographic categories. The following is a strong source for those estimates: https://williamsinstitute.law.ucla.edu/publications/how-many-people-lgbt/

Author Response 9: We have added two population estimates of sexual identity on page 10 of the Results section. See excerpt below:

“While estimates of sexual orientation are not provided by the U.S. Census Bureau, our sample characteristics were similar to those reported by several large studies assessing sexual minority groups (Gates, 2011; Savin-Williams & Vrangalova, 2013).”

Reviewer Comment 10: On page 9, suddenly there are nonbinary participants that were not included in the description of demographic measures – the authors only discuss asking about biological sex.

Author Response 10: We have since listed “gender” as part of our demographic measures. See excerpt below:

“We asked respondents to report their current age, biological sex, gender, race, ethnicity, and sexual orientation.”

Reviewer Comment 11: P17L296: “The number of lifetime partners increased with age and then reached a plateau in 296 older men” – can the authors include a parenthetical here that defines what older men means?

Author Response 11: Thank you for pointing out an area which needed clarification. We have provided greater explanation. See excerpt below:

“The number of lifetime oral and vaginal sex partners increased with age until 60 years old for males and 40 years old for females, with a declination thereafter.”

Reviewer Comment 12: Discussion: P30L488: Please rephrase the following to indicate “intention to BIRTH children”. People can have children (including trans men) in all types of ways that are not restricted by menopause, including surrogacy or adoption. “The frequency of sex may also be influenced by fertility and intention to have children, a process that must occur for women before menopause (typically, before ages 45-55 years old).” 

Author Response 12: This change has been made. See excerpt below:

“The frequency of sex may also be influenced by fertility and intention to birth children, a process that must occur before menopause (typically, before ages 45-55 years old; Centers for Disease Control and Prevention, 2020; Conroy-Beam & Buss, 2019).”

Reviewer Comment 13: I would recommend that the authors pull in Savin-Williams work on “mostly heterosexual” men. Austin has done some work related to “mostly heterosexual” women as well. 

Author Response 13: We have added a reference to Savin-Williams’ work on page 10 of the Results section. See excerpt below:

“While estimates of sexual orientation are not provided by the U.S. Census Bureau, our sample characteristics were similar to those reported by several large studies assessing sexual minority groups (Gates, 2011; Savin-Williams & Vrangalova, 2013).”

---

## [Editor Report · Decision Letter 1]

15 Jul 2021

Age, Sex, and Other Demographic Trends in Sexual Behavior in the United States: Initial Findings of the Sexual Behaviors, Internet Use, and Psychological Adjustment Survey

PONE-D-21-09417R1

Dear Dr. Roberts,

We’re pleased to inform you that your manuscript has been judged scientifically suitable for publication and will be formally accepted for publication once it meets all outstanding technical requirements.

Kind regards,

Andrew R. Dalby, PhD

Academic Editor

PLOS ONE
---

## [Editor Report · Acceptance letter]

29 Jul 2021

PONE-D-21-09417R1 

Age, Sex, and Other Demographic Trends in Sexual Behavior in the United States: Initial Findings of the Sexual Behaviors, Internet Use, and Psychological Adjustment Survey 

Dear Dr. Roberts:

I'm pleased to inform you that your manuscript has been deemed suitable for publication in PLOS ONE. Congratulations! Your manuscript is now with our production department. 

Kind regards, 

on behalf of

Dr. Andrew R. Dalby 

Academic Editor

PLOS ONE